# Identifying Barriers and Facilitators for Home Reconstruction for Prevention of Chagas Disease: An Interview Study in Rural Loja Province, Ecuador

**DOI:** 10.3390/tropicalmed8040228

**Published:** 2023-04-18

**Authors:** Benjamin R. Bates, Majo Carrasco-Tenezaca, Angela M. Mendez-Trivino, Luis E. Mendoza, Claudia Nieto-Sanchez, Esteban G. Baus, Mario J. Grijalva

**Affiliations:** 1School of Communication Studies, Ohio University, Athens, OH 45701, USA; 2Infectious and Tropical Disease Institute, Department of Biomedical Sciences, Heritage College of Osteopathic Medicine, Ohio University, Athens, OH 45701, USAgrijalva@ohio.edu (M.J.G.); 3Center for International Studies, Ohio University, Athens, OH 45701, USA; 4Centro de Investigación para la Salud en America Latina, Escuela de Ciencias Biológicas, Pontificia Universidad Católica del Ecuador, Quito 170530, Ecuador; 5Socio-Ecological Health Research Unit, Department of Public Health, Institute of Tropical Medicine, 2000 Antwerp, Belgium

**Keywords:** Chagas disease, social construction of health, home reconstruction, barriers to change, facilitators of change, interview methods, Ecuador

## Abstract

Background: Chagas disease (CD) is a tropical parasitic disease spread by triatomine bugs, which are bugs that tend to infest precarious housing in rural and impoverished areas. Reducing exposure to the bugs, and thus the parasite they can carry, is essential to preventing CD in these areas. One promising long-term sustainable solution is to reconstruct precarious houses. Implementing home reconstruction requires an understanding of how householders construct barriers and facilitators they might encounter when considering whether to rebuild their homes. Methods: To understand barriers and facilitators to home reconstruction, we performed in-depth qualitative interviews with 33 residents of Canton Calvas, Loja, Ecuador, a high-risk endemic region. Thematic analysis was used to identify these barriers and facilitators. Results: The thematic analysis identified three facilitators (project facilitators, social facilitators, and economic facilitators) and two major barriers (low personal economy and extensive deterioration of existing homes). Conclusions: The study findings provide important loci for assisting community members and for agents of change in home reconstruction projects to prevent CD. Specifically, the project and social facilitators suggest that collective community efforts (*minga*) are more likely to support home reconstruction intentions than individualist efforts, while the barriers suggest that addressing structural issues of economy and affordability are necessary.

## 1. Introduction

Chagas disease (CD) is a neglected yet preventable tropical disease that affects more than 8 million people worldwide [1]. It is mainly found in tropical climates in Latin America; however, due to global migration, the disease is also present in the United States of America and Canada [2], Europe [3], and Japan [4]. In Ecuador, Chagas disease affects more than 200,000 people, and at least 3.8 million—approximately one-quarter of the population—are at risk of infection [5].

*Trypanosoma cruzi* (Kinetoplastida: Trypanosomatidae: Trypanosoma) (Chagas, 1909), the etiological agent of Chagas disease, is transmitted through the excreta (feces and urine) of triatomine insects (Hemiptera: Reduviidae: Triatominae), known in English as ‘kissing bugs,’ in Ecuador as chinchorros, or by many local names in different countries [6]. Although there is evidence that these insects are increasingly found in urban settings in Central and South America [7], more commonly, these insects inhabit poorly constructed homes in rural areas, living in the cracks of the walls, ceilings, and floors of adobe houses [8]. These homes are prevalent in rural Ecuador, and they facilitate infection [9]. *Trypanosoma cruzi* infection occurs by autoinoculation. When the triatomine bug emerges at night to feed, it defecates or urinates as it feeds or immediately thereafter [10,11,12]. People then rub the infected bug feces into the bite or the mucous membranes of the eyes or mouth. The initial symptoms of Chagas disease are like those of a cold, including fever, headache, skin lesions, and swelling. However, over time, *T. cruzi* affects the neurons that innervate the human digestive tract, leading to digestive alterations, and the destruction of cells in the heart leads to cardiac disorders, causing death in many cases [13].

Treatment for Chagas disease remains limited, and currently there is no vaccination. Therefore, it is essential to prevent the bugs from biting people if we want to limit the spread of *T. cruzi*. Since kissing bugs mostly bite people at night, we should protect people while they sleep. The most common way to control bug infestation is using indoor residual insecticide spraying. To be effective, this should be done in a technical manner and using personal protective equipment [14]. However, due to shortcomings of national vector control programs, this is seldom done in most endemic areas of Ecuador [15,16]. Usually, to keep the household free of insects, people spray with the same chemical they use for their crops. This solution, however, is short-term, risks negative health consequences for families due to pesticide exposure, and could lead to the development of insecticide resistance [17].

An alternative solution for primary prevention is improving house infrastructure [18]. Improving a house so that it prevents kissing bugs and other insects from getting into the home has emerged as a potentially sustainable and long-term solution [19]. Infrastructure studies in Latin America have supported the importance of housing infrastructure for the prevention of triatomine entry [20,21]. 

Not every family, however, is able to reconstruct their home. Participating in home reconstruction means, in most cases, needing to leave the home for up to six months, demolishing all or part of the old home, building new adobe blocks, obtaining material from the nearby city, and arranging for help in constructing the new home or the new parts of the home. Although Nieto-Sanchez and her colleagues [22] were able to identify families that could keep their homes free of triatomines, they also concluded that it is necessary to engage households to determine the family’s readiness for home reconstruction. If home reconstruction is a strong and sustainable long-term solution to home infestation, it is essential to identify the facilitators and barriers to participation in home reconstruction identified by community members. Therefore, the present research asks:

RQ1: What are the social constructions of barriers and facilitators articulated in three rural communities in southern Ecuador? 

## 2. Materials and Methods

### 2.1. Setting

Since 2010, the “Healthy Living Initiative” (HLI), a project guided by the Infectious and Tropical Disease Institute at Ohio University (ITDI) and the Center for Research on Health in Latin America at the Pontifical Catholic University of Ecuador (CISeAL), has collaborated with communities in rural Ecuador to ameliorate conditions conducive to the spread of Chagas. Most of HLI’s work takes place in Loja province, Ecuador, which has a high rate of triatomine infestation and, consequently, homes with a high risk for Chagas disease presence [15,23,24]. In particular, the communities of Guara, Chaquizhca, and Bellamaría, located in Cantón Calvas in Loja, were selected as the primary intervention communities for HLI because of the high rate of infestation of triatomines [15,22]. In these three communities, previous research has found household infestation and colonization by *Rhodnius ecuadoriensis* (Lent & León, 1958), *Triatoma carrioni* (Larrouse, 1926), and (to a lesser extent) *Panstrongylus chinai* (Del Ponte, 1929) [15,22]. These three neighboring communities comprise 148 family homes spread through mountainous terrain in Ecuador’s southern highlands (See Figure 1). All three communities have similar characteristics, with most of the inhabitants participating in subsistence farming or day labor. These communities face limited job opportunities, have poor access to health and education services, and are isolated and marginalized from larger markets. Most of the homes in these communities—except those reconstructed with HLI’s assistance—have architectonical, spatial, and structural conditions associated with triatomine infestation.

Because so many homes have features that allow triatomine infestation to occur and thereby raise the risk of infection by *T. cruzi*, HLI seeks to improve the living conditions of communities in southern Ecuador based on three key elements: health promotion, income generation, and infrastructure improvement. The architectural component of the initiative began in 2012 with an assessment of the level of decay present in all the houses of the communities [25]. In the subsequent years, the project improved two houses and built six houses and a community center with the active participation of community members. The techniques used are based on traditional knowledge with technological improvements that result in houses that improve the indoor temperature conditions, are safer, are easier to maintain, and, most importantly, are better able to prevent insect entry and home infestation. 

To achieve the larger objective of the initiative, it is crucial to scale up the housing project. These homes cost, on average, 23,700 USD to rebuild. This need led HLI to involve other institutions (for example, Ecuador’s Ministerio de Desarrollo Urbano y Vivienda (Ministry of Housing and Urban Development)). It also led HLI to rely more on the strengths of the communities and the families the initiative works with [26]. To support HLI’s effort to better involve community members in their own home reconstruction, we needed to assess the facilitators and barriers that potential partner families construct related to the project.

### 2.2. Study Aim

We sought to identify facilitators and barriers to participation in home reconstruction using thematic analysis of interviews with community members in at-risk small communities. Identifying these themes provides important insight for advocates and practitioners of home reconstruction.

### 2.3. Design and Analysis

To explore how community members’ construction of facilitating and limiting factors for their participation in home rebuilding efforts, we used semi-structured in-depth interviews. As part of a larger project investigating factors associated with housing stock, community development, and CD prevention, we conducted 33 interviews with members of the Guara, Chaquizhca, and Bellamaría communities from 1 June to 6 July 2018. 

Because of ITDI’s and CISeAL’s long-term presence in the communities, all members of the three communities have engaged in different levels of participation in the overall HLI. To mitigate against selection bias [27,28], we therefore intentionally approached participants who had informally expressed different levels of involvement and interest in involvement in the home reconstruction component of the intervention, from those not wishing to participate in home reconstruction at all to those who had completed the home reconstruction. As our overall project seeks the scaling up of an intervention, this approach to multiple levels of participation is recommended when interventions are applied to small communities and when the interventions are applied over a long term [27,28].

All participants were adults. Participants were asked to read or have read to them a consent form. Participants gave either signed informed consent or provided a thumbprint to indicate consent. The study was approved by ethics review boards at the Pontifical Catholic University of Ecuador (Protocol # 2018-33-EO) and Ohio University (Protocol # 18-D-97). We conducted the interviews in the households, where the interviewees were predominantly female, and in the field, where mostly men were working. About half of our participants were women (*n* = 16), with men comprising the remainder (*n* = 17). Participants ranged in age from 19 to 78. Educational levels clustered around primary schooling or some secondary schooling, and income levels were generally low. 

Standard interviewing procedures were followed. Community members were interviewed in their homes by Spanish-speaking team members. To control for interviewer bias [27,28], newer members of the project team performed the interviews so that pre-existing relationships with participants or commitments to HLI would be less likely to influence the conduct of the interview. Interviewers were trained in interviewing techniques and used a common interview guide. Each total interview lasted about one hour, with about 20 min concerning issues related to home rebuilding. 

Interviewers employed a common series of questions, which had been developed by the research team and then reviewed and revised by officials from Pontifical Catholic University of Ecuador and validated by local community members. The full interview guide is available: https://osf.io/g2c5s/?view_only=16c6b7ee21734cacb229346da68d0523 (accessed on 11 April 2023). Interviewers were encouraged to include follow-up probes or to revise wordings in questions to seek full exploration of the issues by as many participants as possible. Audio recordings were transcribed in Spanish. Full transcripts were not translated into English; only quotations presented in this manuscript were translated and edited for readability. Each participant was assigned a pseudonym.

Interviews were analyzed using a constant comparative method. While transcribing interviews, the second, third, and fourth authors wrote notations and memos as ideas emerged. By noting potential themes, an emergent reading of the data was allowed. Following transcription, the transcripts were assigned to different readers, and additional themes were identified. After themes were identified, specific instances where interviewees articulated a theme were identified, described, and conceptualized to articulate the significant reasons people gave for being able to or unable to participate in home reconstruction. The major themes were then formed into the findings presented below.

## 3. Results

Participants articulated a variety of facilitators and barriers to participating in home reconstruction. Some of the facilitators were based on ITDI and CISeAL’s long-term presence in the community; trust in both institutions and knowledge of their past successes made it more likely that community members would seek to participate. Other facilitators came from relationships the homeowner had with others, specifically having young people in the home and having built social capital with other community members. Finally, economic characteristics, such as extra agricultural production or access to loans that could be used to finance participation, were seen as facilitators. Economic factors, however, were also articulated as barriers, in that lower levels of personal economy and extensive deterioration of the current home limited participation in the project.

### 3.1. Project Facilitators

One of the most powerful facilitators was that community members believed that ITDI and CISeAL helped community members build better houses. Andrea summarized the implementing partners’ reputations when she said, “Yes, I’ve heard of the initiative. But [they] haven’t come here, to inform us. I have heard of [the initiative] around”. Moreover, what people in the communities had heard and experienced created a positive perception of the two institutions. As Daniela told us, “They have come and we are grateful because, since they started coming, we have seen changes here. They have helped us a lot in things related to the chinchorro”. 

Community members knew that the home rebuilding effort was a long-term and complex project. Santiago outlined his understanding of the project. He said,

This was a health improvement for the population, for the communities. Maybe the people from the University or the people from Ohio, the doctor’s colleagues, have told you something about how it [the project] has been handled, how we have been working, in what way. For me, I would say that the project is a great help.

Santiago recognized that the purpose of home rebuilding was not to give aid to a single family but to improve the larger social whole. He also identified CISeAL, “the people from the University,” and ITDI, “the people from Ohio [University],” as partners with the community. Although the project was most closely identified with “the doctor” (the last author), Santiago saw other members of the project team as colleagues of the doctor and not mere visitors disinvested in the success of the project. These kinds of attitudes led community members to see each visit by team members as contributing to a shared long-term goal. As Viviana put it, “That’s how they are working. I think they are going to do little by little. They cannot help all the houses at once. Then we will also have some hope that we’ll be next”. Gabriel agreed, “it’s a good start, a good help too”.

The primary purpose of home reconstruction was to make homes less susceptible to infestation, yet this was not the main result perceived by the community members. Rather, the homes that were reconstructed became “beautiful”. For instance, Andres told us, “I know very little [about why they want to rebuild the houses]. I know that it is so that the chinchorro does not live there and does not infect people, it is mortal. I know that the houses are more beautiful. The houses look better, and it would be good [to have them] in the whole community. I’ve seen the houses and they are nice”. Cristina Maria agreed, saying, “While walking the path, I have gone to Leonardo’s, and to Juliana Marco’s, and their houses have been made beautiful. Just beautiful”. She continued, “They’re better [houses] than those [traditional houses], not like my house”. Paula echoed these remarks: “The houses are beautiful”. Beyond this general judgment of beauty, community members gave reasons for this judgment. Lore said, “The houses are very nice and well covered. No animals enter them, unlike here”. Isa noted, “It’s great because everything is clean there. Everything is in its places, the bedrooms, water, bathroom, uncluttered walls”. Additionally, Edison stated, “I think it’s a super-good project. I have even worked on the houses. I have seen how they do the processes to make this kind of magnificent place. The material, even the adobe. The house has a freshness”. Whether the house was beautiful because there were no animals in it, was better organized, or was airier, community members agreed that if your house was reconstructed in partnership with ITDI and CISeAL, it would be a better house.

Community members also agreed that the house would also be healthier and safer. Although Andres had a very general overview of the dangers implied in having chinchorros inside the house, other community members thought that excluding all insects was a good idea. For example, Viviana said, “The houses that are being built are beautiful, healthy homes because they do not get flies, mosquitoes, or chinchorros”. Insects were not the only thing excluded. Fiorella reported, “Of course, we know about the support that they are giving in these projects. They know how to build houses to make them safe. Insects do not enter; neither do mosquitoes or the rats. In houses where they have holes, rats abound”. The houses were also safer because they were made of good materials. Daniela thought the adobe was better because it was more structurally sound. She said, “We would live better in an improved house. Even if it is made of brick, or if we change the floor, that changes things. The wind blows outside at times and even if you sweep everything away, it [dust] quickly fall down”.

When asking individuals to rebuild their homes, both trust in the overall project and a good reputation for accomplishing the goals of the project are central enabling factors. Because the community trusted ITDI and CISeAL, a trust built up over more than a decade of interaction, community members were likely to trust the organizations when they suggested a major structural change. Likewise, when community members perceived that the home they would have at the end of the project was more beautiful, healthier, and safer, they would be more likely to accept participation in the project.

### 3.2. Social Facilitators

Social facilitators included the relationships that community members had with other members of their community. These relationships provided social resources that could be used to support home reconstruction.

Young people in the home as facilitator. The presence of young people in the home allowed community members to be more engaged. Although smaller families are sometimes seen as facilitators of social and economic development [29], the presence of young people who could work—either for funds from employment outside the home or helping in the physical reconstruction process—allowed, in this case, greater participation in home rebuilding. For example, when Manuela was directly asked, “if you came to work with us in the houses, do you think there are people who would help you with the work?” Manuela replied, “Yes, the boy that I have [can work]”. Her boy, her 24-year-old son, was available as manual labor and could help Manuela rebuild her home. Cristina Maria had an even greater labor force; she proudly told us, “All of my children are grown and hardworking. Only one is too young”.

Having strong children who could work would make participation easier. Fiorella was asked if she thought there were people in her home who could help. She replied, “Yes, there must be about ten people who can [help]”. It was important that these children also be old enough to help but not yet have families of their own to take care of. Santiago, when asked if his four children could work, said, “Those who can work [can help]. Two, not yet; they are minors”. The minor children may not be strong enough to help, or they may need to go to school. Alternatively, Franklin said that, until recently, he had two sons who could help, but one moved to a coastal city. “I had two, but now I have one [who can help]; the other cannot”. Sometimes young members of the family migrate to bigger cities to work, and although they visit their families, they are not considered part of the family workforce.

Social capital as a facilitator. When asked directly whom they would turn to if they needed help in rebuilding their homes, community members who had a strong social network were more likely to feel able to participate. An extended conversation with Lisette shows the importance of being able to turn to a range of people. When asked how her family was able to gain help, she told us:

Lisette: Yes, the two of us worked [building our house]. Sometimes we asked people. That’s all we did, but only for the roof. We asked for people to help with the roof. We did it both of us together. Of course with people from the community.

Interviewer: So do you believe that the people of the community would help you, like friends or family? Lisette: Yes. Because we never ask for help. We ask Mir for help. Mir is always the one who helps. Abelardo also helps us sometimes with his car. Those two are the most helpful. The rest do not help.

Interviewer: If you need money who do you ask for it?

Lisette: We asked one of Nixon’s [the head of household] brothers-in-law, and we have a loan in the bank.

In this conversation, Lisette stated that when they reconstructed their home, they did not usually need to get help from others. Their roof, however, was particularly difficult—both financially and physically—to erect. Their connections with other people in the community, with the family, and with the bank were all brought to bear to make the roof possible. Community members could provide labor and other resources that enabled home rebuilding. For example, Angel was asked if he had someone to help him rebuild his home, and Angel replied, “No, there isn’t, but if you pay someone they will help”. Angel’s response did not mean that there were no other members of the community who could work. Rather, local values distinguish between selfless help and paid help. When working on a home, which is a family asset, selfless help is unavailable, but one can pay for assistance in homebuilding.

The social capital ties also reveal that access to financial support from the community is tied to people’s ability to assist in home reconstruction. Generally, helping to reconstruct a home takes away from time to work for wages or to tend crops. Therefore, being able to pay these community members is important in strengthening these ties. Gabriel, for example, said that if he needed help, he could turn to “the community, to friends for a day or two. I could help [them] too,” but that they would be more likely to help if you could pay them. Some members, like Gabriel, also talked about exchange in workdays; this is seen as a facilitator and an alternative to paying in cash for work. Lore said that she hasn’t been able to do home projects not only due to the lack of money but also “because we are alone. How could we do it alone, and without money to be able to do something?” Here, Lore explicitly ties together financial resources and being able to get other people to assist in home reconstruction. Isa—in her interview—agreed, saying, “yes, yes there are people [that can help with the construction work], but you need money to pay them. There are people that can help, but there are no jobs”.

These two sources of social facilitators are significant. If larger families are able to help community members rebuild their homes, provided that the children are old and strong enough to assist, this represents a powerful source of free labor. These kinds of families, however, assume that there are adequate resources within the individual home to support family and individual development. If there are too few children, or the children are not old enough, ties into the community can provide additional labor. This turn to the community, however, also assumes that the original community member is able to afford to pay the other community members in cash or workdays. These community members are not volunteers; they are, essentially, day laborers.

### 3.3. Economic Facilitators

This need to pay laborers indicates that, beyond social connections, access to additional money is essential to home rebuilding. Interviewees indicated two sources of funds that they could seek out: excess agricultural production or loans.

Extra agricultural production as facilitator. Most community members practice subsistence agriculture. In some cases, however, families have access to sufficient land to grow additional crops that they can take to market. These funds, then, can be used for home reconstruction.

Several of the participants remarked that they are sometimes able to grow extra food to sell. For example, Andres stated, “My wife, my children and I work in the garden. We grow winter corn, also cilantro, things like that. When my children live in Cariamanga for their studies, they take it [the produce] to sell, in that way they have [money for bus] tickets” to get to the town. Andres was able to piggyback on top of an expense the family already had to try to turn it into a small profit center. Because they had already had the cost of sending their children to town each week, they asked their children to take with them fruits and herbs that could be sold. Heidy, the owner of a larger garden, told us, “I have a garden of 4 hectares. My son works there. We grow yucca, banana, sugar cane and zarandaja (beans). Mango, orange, and guava. Passion fruit, lemon, tangerine”. When asked if she brought this to market, Heidy said that the garden provided financially “with the yucca and the zarandaja. At the fair the mango, the zarandaja or the yucca are sold”. Although the family grew diverse crops, the excess production was in yucca and beans; the family consumed the rest. Similar choices were made in other families. Isa said, “[I sell] at the fair. Right now we are harvesting oranges, lemon, sour oranges, tangerines. Tomorrow, I’m going to take out yucca and make some panela (pressed sugar cane). We also sell pigs and chickens”. Adriana also stated, “We have animals to sell, but very few because they die. We have pigs, chickens”. Finally, Lore told us, “We have a tiny garden and also breed animals, calves, piglets”. Although there were sometimes extra crops or animals to sell, the season, the family’s need to eat, and the size of outputs all constrained the ability to take goods to market.

Even when families sell goods, the profit earned is not huge. The most successful family taking goods to market, represented by Lisette, told us, “I take papaya, banana and yucca and the animals to sell. I would sell the pigs... Let’s see, now that I managed to sell everything, I made 800 USD. This is what we get every six months, or sometimes in the whole year that we get this amount”. Although Lisette’s family was having an extraordinarily productive year, she recognized that it was probable that other years would be less productive. Even if she were to set aside the full amount of 800 USD per year, it would require 3 years to meet the family contribution level for reconstruction sought by ITDI and CISeAL. Andrea’s family was the next most successful at the market. She said,

When we have a little extra, we sell it. You don’t get that much either. Sometimes you earn about 300 or 400 USD a year… Sometimes [we sell] pigs, goats, yes. We don’t sell cattle. A pig costs from 50 to about 150 USD. That’s what I raise, but then some [families] raise more. A goat, that’s worth around 30 to 70 USD.

Andrea’s family’s production was less than half that of Lisette’s family’s. Additionally, even though they were relatively successful, it would still take 5 years to raise the necessary funds. Other families were more circumspect in telling us about how much profit there was from market activities, but it seemed that this source of funds was unlikely to grow rapidly, was unsteady, and is insufficient for enabling participation in home reconstruction.

Access to loans as facilitator. Other families indicated that if they needed additional funds, they would seek out loans from banks. This access, while a facilitator for many, was also very constrained. Several community members reported that they had thought about accessing loans, but that the repayment conditions were difficult. That is, the banking system was functionally unavailable to many community members. Lisette told us that she had once tried to get a loan, but the bank asked for the title of the land. When we asked how she would repay a loan to buy seed corn back, she answered, “From the same corn, but also the yucca or from some animals too”. When Lisette’s family asked for the loan, the bank would only lend if they agreed to collateralize their corn crop and their other agricultural production. This collateral risk was, for Lisette’s family, too high. Additionally, should the bank seize their crops, repayment “is difficult because sometimes there is no income, there is no money to pay [the interest]. Loans for maintaining the household, even if the situation becomes stable, are a constant concern. If you get a loan is because you can look for other jobs, other means to pay that” loan back.

Loans were available, but not on terms as good as they used to be. Angel told us that he borrowed from the bank “years ago, but then no more. The interest is expensive and, most of all, you have to pay every month and there is no money to pay it. Sometimes you get money to buy food but then you don’t have money to pay the monthly fee”. Angel saw the tradeoff between feeding the family or paying the loan as too high a risk. Paula agreed—she would ask a bank for a loan, except “I do not have chances to pay. I am afraid to take out a loan because I do not know how I would pay” the bank back.

Some community members mentioned that they would ask their children working outside the communities for money before considering the bank. Lore told us, “well I would ask my sons for help. I have been telling them that they have to fix my house. I am scared that it [the house] crushes me because the beams are so old”. Similarly, Gabriel said, “I won’t ask them [sons living in the city] because they are renting a room, they barely can support their own families… Maybe I would ask Solange [his daughter who lives abroad]”. Young community members who live away can be an asset for reconstruction. Even if they cannot contribute with physical work, they represent a valid alternative to bank loans. Other alternatives to bank loans included accessing the government welfare plan loans. Ana is enrolled in the welfare program but hasn’t received her monthly payment because she got a loan the previous year: “I got a loan last year, to pay the monthly fees we sold some pigs. We are paying it”. If families get a loan from the welfare program, it will be automatically charged to their monthly payments, meaning that they won’t get that amount until the total of the loan is covered.

### 3.4. Barriers

The personal and structural factors that make loans practically inaccessible show that the challenges in rural Ecuador may not be addressed well by agents from urban areas. The barriers to home reconstruction specifically articulated by members of these communities reflect how essential it is to understand living conditions in these communities. The community members told us that the economic challenges and the existing conditions in their homes made it difficult to contemplate rebuilding their homes.

Low personal economy as barrier. Many of the families stated that they were too poor to invest in their homes. Gary told us that he knew he needed to reconstruct his home but could not afford it financially. He said, “Of course that we need is to fix the house, to have a bathroom that is in good condition and a kitchenette is missing. Now we have running water, we have electricity. The need for money has us controlled, it is not enough”. When asked directly, “why have you not made the changes you want in the house,” community member responses followed a pattern. Lore said, “I do not have the money. The money, I do not have it”. Isa said, “Because we lack economic resources”. Santiago agreed, “What is missing? It is a lack of financial resources because in some way, as you know with resources, with money we can make changes”. Viviana told us, “Lack of money, not enough [money]”, and Angela said, “For lack of money”. “We need the money,” concluded Daniela.

Lack of money is not a given but rather emerges from the challenges of living in a rural and underserved community. When education is needed, it cannot be acquired locally, and seeking educational or health services can hurt these families economically. Andres said he could not save for rebuilding his home “because I already have three children in high school. My income no longer supports us. What is earned is very little”. Manuela agreed, saying, “It is not possible to [spend on rebuilding]. We lack the money. With the children that we have had, we have lacked the money because we do not have it”. Children, when of working age, could assist in reconstructing the home, but if they were young enough to need to go to school, they added difficult expenses for the family.

Even when families had begun to save, a single illness could erase their life savings. Cristina Maria reports that she has not been able to make changes in her house “because of poverty. My husband fell ill, he feels sick”. Lisette’s story is similar: “We thought about making those changes. Then a son got sick and we had expenses, and as long as we do not see the end of it [the illness], we cannot do anything”. Paula also told us she could not improve her house “because I have not had money and my daughter’s illness complicated everything”. She explained that her daughter “had nervous system problems and she was close to death. My son took her to get medical attention”. These challenges present a double challenge to the families; not only do they need to spend money on doctors and medications, but they have also lost a productive earner for the family over the course of their illness.

Extensive deterioration of existing home as barrier. These challenges to earning and then saving money for home reconstruction often led families to defer maintenance. Community members told us that they had delayed reconstruction for so long that it simply was not worth rebuilding. In addition, as part of ITDI and CISeAL’s efforts, the old home must be removed before the new one can be constructed or, if there is enough area in the family’s plot of land, they could stay in the original house until the new one is ready.

Manuela told us that it was not necessary to renovate her home. When asked what she would need to consider rebuilding her home, she said,

Nothing. What I do not want is to take this house down, because we have lived here so long. We have sacrificed to do it. Also because if one of my children has nowhere to live, he at least can live [here]. We are many and we are poor. And we don’t have the chance to build a house at this time.

For Manuela, the home, as poor as it was, was an inheritable asset. She told us that even if she and her husband could get a new house, she would like to keep her current house for her daughter, who recently had her own children. This shows that even if they consider their house to be old or in need of improvement, they wouldn’t consider bringing it down.

Indeed, the idea that homes would need to be torn down before new homes could be rebuilt created a significant barrier to participation. When we asked what it would mean if we told them that rebuilding their home would require demolition of the old one, Jael reported that, physically, his family could participate “because we would take out everything that is more or less worth something and [a new house] could be built outside without problems”. Similarly, Ana admitted that she might participate in rebuilding, yet she said, “it would be a lot of work tearing down the [current] house and building it up again”. Some participants discussed the disruption that tearing down the house implied. The disruption was that when you reconstruct a house, you are also tearing down a place that people think of as a home. Additionally, above concerns about losing this home, Nohely expressed that some people might not participate because they would not be able to have a home ever again. When asked if she would agree to tear down her house, she said,

No. I do not like this house, but I would want build a new one first. I have land over there. I don’t want to be like Doña Kora that tore down her house and then she was living from [temporary] house to [temporary] house.

Even though Nohely was dissatisfied with her house, it was still her home. Because she had more land to build on, unlike a member of the community she had heard about, she did not think it risky for her to tear her house down if she had a new one built. She understood, however, why others would think it risky because they did not have the space to build a new house while living in their old one.

## 4. Discussion

Members of the communities of Guara, Chaquizhca, and Bellamaría identified factors they constructed as enabling them to participate in home reconstruction. They also constructed significant barriers. Each facilitator and barrier provides important insight into identifying families that view themselves as ready for home reconstruction. The community member’s constructions of resources and community empowerment as significant factors are similar to those identified in previous research on home reconstruction [20,21]. The community members, however, add an additional significant facilitator that is little mentioned in previous work: project reputation and trust.

The project facilitator—the positive reputation enjoyed by HLI—is a significant asset for home reconstruction efforts. Asking a family to destroy their old home so that a new one can be built requires a great deal of trust. With time, the initiative has identified ways of negotiating the demolition of the house to make it less disruptive for the families. There is, unfortunately, a long history of neocolonial exploitation of communities in the global south, which may make such trust difficult [30]. In addition, at the local level, there is a distrustful relationship between the government and the impoverished rural communities, partly due to populist politics, neoliberal policies, and a low level of enforcement of citizenship rights [31,32]. These policy choices have created a sense of abandonment among rural populations, who face limited access to education and health services—services that are usually located in the urban or periurban areas. Even though the national government has invested some of Ecuador’s oil revenues in development in the study area, this investment has mainly been for road infrastructure rather than health and education facilities.

Considering that trust has been identified as a fundamental element in fostering community involvement in vector control programs [33], ITDI’s and CISeAL’s longstanding work with these three rural communities represents an important asset for the program. Moreover, establishing trust helps to socially construct family-level, multi-person perceptions that involvement in the intervention is helpful to the family, confirming previous findings at the individual-intervention level in the social construction of health literature [34,35]. Because HLI did not immediately begin reconstructing homes, but instead introduced home reconstruction after multiple years of careful, methodical, and—most of all—respectful work with members of the communities, this initial investment appears to have paid off with community trust and perceived closeness. Other organizations seeking to implement high-impact projects may wish to model HLI’s approach of creating a positive reputation, ongoing communication, and a favorable environment for their organization through smaller-scale and immediately impactful efforts. Investing in such efforts as major interventions take place can help to keep building such assets while sustaining the transparency perceived by community members. Similar to other development interventions, there is substantial space for unintended consequences if this investment in relationships is absent [32,36]. By accepting this intervention in their homes, these families allowed HLI to dismantle their most valuable asset because they were promised a safer home. In this context, it was important that representatives of HLI demonstrate that the families possessed knowledge and, ultimately, had control over the intervention. As such, intervention and research should be understood as an ongoing learning experience that is constantly reframed by what communities and staff learn about each other over time and how they construct the meaning of home reconstruction together. This finding should be tempered, however, by the potential for selection and interviewer biases. It is possible that interviewees engaged in desirable responding to obtain additional resources from HLI or that interviewers unintentionally prompted responses that reported trust. Because of the long-term engagement across the community, although best practices were followed to control these biases, the potential for these biases to emerge remains possible [27,28].

The two social facilitators identified by community members were the presence of young people who could participate in home reconstruction and strong social networks that could be used to recruit friends and neighbors to support the process. These social facilitators highlight that home reconstruction is not an individual effort; home reconstruction is a familial and community effort. This community work may be a cultural manifestation of ‘*minga’* practice, an Andean conception of collaborative work made by native communities before colonization and maintained to the present [37]. This collectivism may require home reconstruction efforts to involve multiple members of the community and to advance the collective good instead of individual goods. HLI’s previous investments in community education and health promotion [18,26,38], as well as their construction of a community center, may have built on *minga* orientations. Nevertheless, larger interventions should not assume that community work will be provided for free. Labor is a significant main resource for every family, and unpaid collaboration may become seen as a loss of that resource. If a family requests this kind of collaboration from their neighbors, they will, in the present, be asked to provide food for all workers and, in the future, perform work for their neighbors as compensation. Therefore, to sustain and expand previous efforts, HLI may wish to attempt to reframe home reconstruction as *minga* efforts. While keeping close relationships with individual householders, project implementers can build upon the social facilitators identified in this study to foster the existing interest and involvement in home reconstruction in the communities at large.

The two economic facilitators identified by community members were both related to obtaining funding for home reconstruction. Just as individuals participate in socially constructing the access and affordability of individual-level interventions [34,39], advocates for family-level, multi-person interventions must address social constructions of access and affordability. The families involved in this study noted that if there was excess agricultural production to sell or if the family could access a bank loan, they were more likely to say that they could raise the needed funds. These facilitators were directly related to the two barriers; low personal economy and extensive deterioration of the home limited their perceived ability to reconstruct their home. Previous home reconstruction efforts have reported on the important impact that availability of funding has on individual-level decision-making about home reconstruction [40,41]. Rebuilding homes is very expensive for community members; the full cost of reconstructing a home is about 25,000 USD, which could represent the lifetime earnings of a family. Subsidies from ITDI, CISeAL, and their partners have reduced costs to the families to between 2000 and 2500 USD. Although familial investment in their own homes may foster ownership, it means that some families must commit years of income and that some families will find participation unaffordable [25]. Addressing the community members’ constructions of their current social state may help resolve this gap. Promoting greater agricultural production may make homes more attainable. Working to make loan schemes more compatible with these communities’ needs may also help. Additionally, more affordable forms of protecting homes from bugs could be implemented as a stopgap before scaling up infrastructure improvements.

The interviews allowed us to identify the barriers that the families construct related to HLI’s home reconstruction projects. Because the study participants were involved in ten years of previous projects, their concerns might not be the same as other communities. The results, therefore, cannot simply be generalized to other communities, particularly given the relationship that has developed between the ITDI and CISeAL staff and the members of the communities. Nonetheless, the knowledge gained from these interviews can be a basis for understanding and addressing family-level facilitators and barriers and for developing construction plans appropriate for each family. Future studies employing quantitative methodologies could assess associations between these barriers and facilitators and actual participation in home reconstruction, or could explore the relative prevalence of these factors in communities that have not yet been engaged by HLI.

Finally, this study highlights possibilities for future research. The three communities this study worked with are similar and adjacent to one another. There may, however, be differences among communities. For instance, houses in Guara are more geographically dispersed, which may make it more difficult to strengthen the social fabric of the community. Most of the families of Chaquizhca are interrelated, which may facilitate *minga* work; additionally, Chaquizhca and Bellamaria have recently constructed their own water systems, which may have developed a sense of community. These features may influence the capacity of families to reconstruct their houses and may change the balance between individual and collective efforts. In addition, house (re)construction offers the opportunity to assess the effectiveness of this approach in preventing (re)infestation of homes. As indicated earlier, three different species of triatomines have been found in domiciles in these communities (*R. ecuadoriensis*, *T. carrioni*, and *P. chinai*). As reported in our earlier work [15,22], these species infest homes at different rates. We do not know the specific impact of this intervention on the three species we have encountered for likelihood of home (re)infestation following home reconstruction. Our anecdotal observation is that none of the eight homes that have been rebuilt have become infested. We are reluctant to report additional findings about home (re)infestation here because systemic investigations of homes that have and have not been reconstructed is a project we intend to conduct in July 2023 and July 2025. These variations among communities and insect species, along with other differences that may make a difference, may allow this research to serve as a base for studies on affordable housing and how the construction process impacts the economy and general living standard of families involved in rural communities impacted by Chagas disease.

## 5. Conclusions

Home reconstruction is a long-term and sustainable solution to preventing the transmission of Chagas disease in rural Ecuador. Home reconstruction, however, requires more than one person to accept a family-level, multi-person intervention that addresses deindividuated causal factors. Accepting this solution requires us to identify and activate facilitating factors that encourage participation in home reconstruction and to identify barriers that can be overcome. The barriers to participation were based on economic conditions: poverty and precarious housing. This study provides practitioners guidance in leveraging project, social, and economic facilitators to overcome these barriers. Specifically, practitioners should seek to expand *minga* efforts. Practitioners should not seek to rebuild homes for community members. Instead, practitioners should carefully build authentic relationships with community members and seek to identify existing networks among community members to encourage collective and shared participation in home reconstruction.

## Figures and Tables

**Figure 1 tropicalmed-08-00228-f001:**
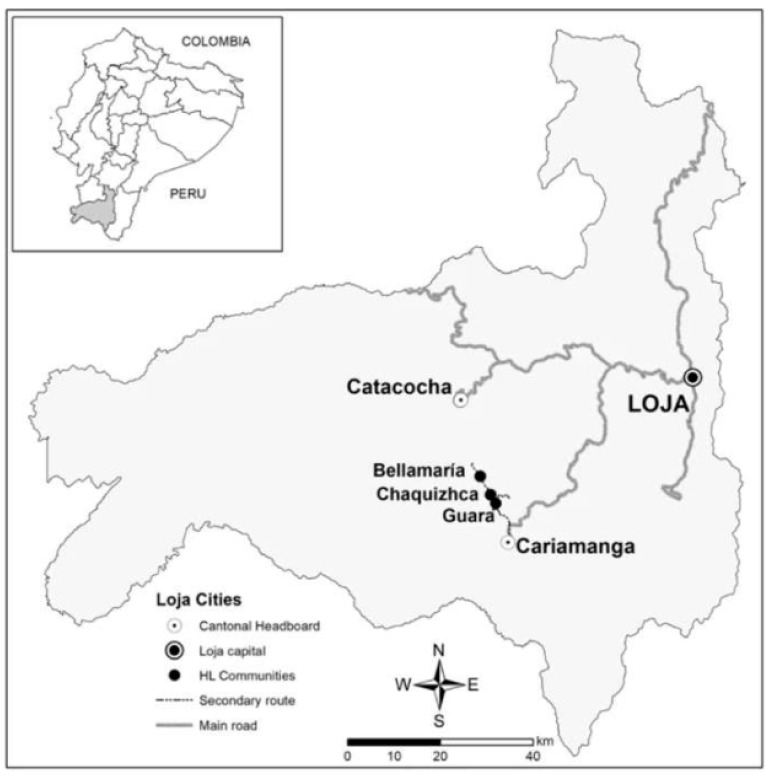
Map of the three communities in relation to Loja province.

## Data Availability

The datasets analyzed during the current study are not publicly available because they contain information that could identify the individuals interviewed but are available from the corresponding author on reasonable request.

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
