# Peer review of "Identifying Barriers and Facilitators for Home Reconstruction for Prevention of Chagas Disease: An Interview Study in Rural Loja Province, Ecuador"

_tropicalmed, 2023, doi:10.3390/tropicalmed8040228_

Round 1

Reviewer 1 Report

The manuscript presents relevant information about the reality of families located in rural areas of Ecuador and highlights the main factors that can facilitate or hinder the completion of home reconstruction.

Below are some suggestions/corrections needed:

Introduction

The first sentence (lines 36 to 37) has no reference.

In line 42, in the first mention of a species, it is important to include the taxonomic classification of the species, for example: “Trypanosoma cruzi (Kinetoplastida: Trypanosomatidae)”.

Line 42: In addition to feces, triatomine urine can also transmit T. cruzi (https://www.who.int/news-room/fact-sheets/detail/chagas-disease-(american-trypanosomiasis)

Line 43: I recommend including the taxonomic classification after “triatomines insects”, for example: “of triatomine insects (Hemiptera: Reduviidae: Triatominae)”.

Line 43: The word “chinchorros” is not formatted correctly.

Line 46: When starting the sentence with the name of a species, it should not be abbreviated, even if it has already appeared before in the text.

Line 53: I believe that “producing” could be replaced with “causing”.

Methodology

In section 2.2, line 113, the currency referring to the cost must be presented. The same must be done for the other mentions of monetary values, for a better understanding of the reader.

In section 2.3, I was unable to access the questionnaire with the questions asked. I can open the link, but when I click on the document, no information appears. Therefore, it is very important that this be revised.

Although it is possible to evaluate the responses of community members, satisfying the objectives of the manuscript, I believe that the use of multiple choice questionnaires (which could be used in conjunction with interviews) would generate quantitative data so that other analyzes could be performed (correlations, comparisons between locations, among others). Thus, I believe that this can serve as a suggestion for other studies that the authors carry out.

Results

In the results section, there are some text formatting errors:

The paragraph between lines 177 and 181 has a formatting error.

Between line 292 and 293 there are also words with a formatting error.

Line 334 there is a duplicate “the”.

The paragraphs between lines 344 and 354 also have formatting errors.

The paragraph between lines 458 and 461 also has formatting errors.

Discussion

The title of the “Discussion” section (line 467) is not formatted correctly.

Abbreviations

I believe that T. cruzi as an abbreviation is redundant, since the abbreviation of the name of a species already mentioned earlier in the text is already expected. However, if the authors consider this inclusion important, it must be presented in italics (line 592), respecting the zoological nomenclatural code.

References

Some references need to be revised, since the journals mentioned are presented, sometimes in abbreviated form, sometimes with the full name. They must follow a single pattern

The DOI of reference 25 is presented as a link, while in the other references only the number is presented. So, you could change it to just the number in reference 25.

Author Response

We thank the reviewer for their comments.

We report their original comments in plain text and our response in italics immediately following.

Introduction

The first sentence (lines 36 to 37) has no reference.

We have added a reference.

In line 42, in the first mention of a species, it is important to include the taxonomic classification of the species, for example: “Trypanosoma cruzi (Kinetoplastida: Trypanosomatidae)”.

Thank you for this recommendation. We have added the fuller taxonomic classification.

Line 42: In addition to feces, triatomine urine can also transmit T. cruzi (https://www.who.int/news-room/fact-sheets/detail/chagas-disease-(american-trypanosomiasis)

We agree. We have added that both forms of excreta can transmit this parasite.

Line 43: I recommend including the taxonomic classification after “triatomines insects”, for example: “of triatomine insects (Hemiptera: Reduviidae: Triatominae)”.

We have added this as recommended..

Line 43: The word “chinchorros” is not formatted correctly.

We have corrected this formatting.

Line 46: When starting the sentence with the name of a species, it should not be abbreviated, even if it has already appeared before in the text.

We have corrected this formatting.

Line 53: I believe that “producing” could be replaced with “causing”.

 We have changed this word.

Methodology

In section 2.2, line 113, the currency referring to the cost must be presented. The same must be done for the other mentions of monetary values, for a better understanding of the reader.

We have clarified that all monetary values are in USD, the currency used in Ecuador.

In section 2.3, I was unable to access the questionnaire with the questions asked. I can open the link, but when I click on the document, no information appears. Therefore, it is very important that this be revised.

We have re-uploaded the file so that the interview guide can be accessed.

Although it is possible to evaluate the responses of community members, satisfying the objectives of the manuscript, I believe that the use of multiple choice questionnaires (which could be used in conjunction with interviews) would generate quantitative data so that other analyzes could be performed (correlations, comparisons between locations, among others). Thus, I believe that this can serve as a suggestion for other studies that the authors carry out.

We agree. This is part of work that will be performed in July of 2023 using a different methodology from the one reported in the present manuscript. We have added this potential direction for future research to the discussion (around line 560).

Results

In the results section, there are some text formatting errors:

Thank for noticing these. We are unsure how these formatting errors came in. We have adjusted all block quotations to be indented from the left.

The paragraph between lines 177 and 181 has a formatting error.

We have corrected this formatting.

Between line 292 and 293 there are also words with a formatting error.

We have corrected this formatting.

Line 334 there is a duplicate “the”.

We have removed this duplicate.

The paragraphs between lines 344 and 354 also have formatting errors.

We have corrected this formatting.

The paragraph between lines 458 and 461 also has formatting errors.

We have corrected this formatting.

Discussion

The title of the “Discussion” section (line 467) is not formatted correctly.

Thank you for noticing this. We have adjusted the formatting of this section title.

Abbreviations

I believe that T. cruzi as an abbreviation is redundant, since the abbreviation of the name of a species already mentioned earlier in the text is already expected. However, if the authors consider this inclusion important, it must be presented in italics (line 592), respecting the zoological nomenclatural code.

We have removed this from the abbreviations list.

References

Some references need to be revised, since the journals mentioned are presented, sometimes in abbreviated form, sometimes with the full name. They must follow a single pattern

Thank you for this comment. Following the MDPI style guidelines, we have journals are abbreviated according to ISO 4 rules in the ISSN Center's List of Title Word Abbreviations or CAS's Core Journals. For all journals not listed, we have left the entire title as advised by MDPI. The Editorial Office will abbreviate those journal titles appropriately, as indicated at https://www.mdpi.com/authors/references.

The DOI of reference 25 is presented as a link, while in the other references only the number is presented. So, you could change it to just the number in reference 25.

Thank you for noticing this. We have removed the hyperlink.

Reviewer 2 Report

The case for housing improvements as a means to Chagas control and eventually elimination is well-made and the housing project referred to in this paper is a very interesting one. The main concerns I have around the methodology are the potential for selection bias and interviewer bias. The paper presents the interviewees as having a wholly positive view of the project, but what about the people who declined to participate? 

More specific comments:

1. The information about the housing project (currently in 2.2. Setting) needs to come before the aim. And further information is needed about what you mean by 'participation in home reconstruction' as this is referred to many times in the paper without prior definition.

2. Study aim states - 'we sough to identify facilitators and barriers to participation...' - clarify what you mean by participation. I assume you mean some form of home reconstruction however, as above, what this entails isn't defined anywhere before the aim. 

3. Power dynamics and the potential for related interviewer bias should be acknowledged and discussed. Were participants just reporting a positive view of the housing project because they were speaking to a representative of it? And were they trying to 'say the right thing' to get additional funds or support? Some of the quotes might suggest this, such as "we will also have some hope that we'll be next" (line 190-191). 

4. As the interviewers appear to be from the housing project, the potential for selection bias is significant in this study. 

5. Some of the quotes suggest leading questions asked by the interviewers. E.g. line 242. Did the interviewers have any experience or training in qualitative/interview methods?

6. Can you comment on how transferable your findings are likely to be to other communities in endemic regions of Latin America considering community-level home reconstruction projects?

Author Response

We thank the reviewer for their comments.

We report their original comments in plain text and our response in italics immediately following.

The case for housing improvements as a means to Chagas control and eventually elimination is well-made and the housing project referred to in this paper is a very interesting one. The main concerns I have around the methodology are the potential for selection bias and interviewer bias. The paper presents the interviewees as having a wholly positive view of the project, but what about the people who declined to participate? 

We agree that selection bias and interview bias is a concern. We believe, however, that it is significant to note that there are two forms of “declining to participate.” One form is declining to participate in the interviews and the other is in declining to participate in home reconstruction. All of the individuals who were approached for interviews agreed to participate; none declined.  

It is significant to note that ITDI and CISeAL have worked with these three communities for more than 10 years in the Healthy Living Imitative, and all families that live in these communities have been engaged by this project. Because this project seeks the scaling up of an intervention, our selection of participants sought to interview individuals who had different levels of involvement in the home reconstruction intervention, from those not wishing to participate in home reconstruction at all, like Manuela, or who had additional conditions that needed to be met before they would consider being involved in the intervention, like Ana to those who were seeking money to be able to participate once they had raised funds, like Andres and Ariana, to those who had completed the home reconstruction process, like Lisette and Nixon. This approach to multiple levels of participation to mitigate selection bias is recommend in best practices in implementation science when interventions are applied to small communities and when they are applied over a long term. We have, in the main body of the paper explained these attempts to mitigate selection bias and provided appropriate references, specifically the National Cancer Institute Qualitative Methods in Implementation Science guidelines and the Administration on Children, Youth & Families’ Consideration for Federal Staff guidance.

We have also explained how we tried to control interviewer bias, specifically by having interviews conducted by newer members of the project team who had not themselves built close relationships with the members of the community and the use of and training in a common interviewing guide.

More specific comments:

  1. The information about the housing project (currently in 2.2. Setting) needs to come before the aim. And further information is needed about what you mean by 'participation in home reconstruction' as this is referred to many times in the paper without prior definition.

We have re-ordered these sections as advised.

  1. Study aim states - 'we sought to identify facilitators and barriers to participation...' - clarify what you mean by participation. I assume you mean some form of home reconstruction however, as above, what this entails isn't defined anywhere before the aim. 

We have added additional information on what “participation” in home reconstruction entails on lines 71-75 of the manuscript, before the study setting.

  1. Power dynamics and the potential for related interviewer bias should be acknowledged and discussed. Were participants just reporting a positive view of the housing project because they were speaking to a representative of it? And were they trying to 'say the right thing' to get additional funds or support? Some of the quotes might suggest this, such as "we will also have some hope that we'll be next" (line 190-191). 

We agree that this is a potential bias, however we think that the more straightforward answer (that these families want to participate in home rebuilding) is more likely and simpler. We believe that this is accounted for as described above and in the addition of directions for future research.

  1. As the interviewers appear to be from the housing project, the potential for selection bias is significant in this study. 

We have addressed this as indicated above.

  1. Some of the quotes suggest leading questions asked by the interviewers. E.g. line 242. Did the interviewers have any experience or training in qualitative/interview methods?

We have addressed this as indicated above. The use of a shared interview guide and training should mitigate this.

  1. Can you comment on how transferable your findings are likely to be to other communities in endemic regions of Latin America considering community-level home reconstruction projects?

We have expanded on this in the discussion.

Reviewer 3 Report

Minor modifications to comments in the text. Congratulations on the quality of the work.

Author Response

We thank the reviewer for their comments.

This reviewer’s comments were embedded in a marked-up pdf file.

We report our adaptations.

We have corrected the formatting when fonts or spacing changed.

We have added the reference about triatomine bugs also inhabiting urban centers.

WE have changed the colloquial reference from kissing bugs to triatomines.

Although we were unable to insert it into the main text, we have provided to MDPI a map of the region that shows the study site. This can be inserted at production if desired.

Reviewer 4 Report

In general is a good paper, with innovative approach. However, the entomological part is missing. The authors did not mention what are the most common/important Triatomines in the country; moreover what are the important species in the area of study. Although we know triatomines are vectos, every genera and even species have some differences in the behavior and incidence in human habitations. Therefore, the entomological component should be added and discussed. Wil the measures taken in this case will affect all the species/genera. Will be the same for all in the triatominae in the area? etc. Are there some species for which changes in construction may not affect that much?

Author Response

We thank the reviewer for their comments.

We report their original comments in plain text and our response in italics immediately following.

In general is a good paper, with innovative approach. However, the entomological part is missing. The authors did not mention what are the most common/important Triatomines in the country; moreover what are the important species in the area of study. Although we know triatomines are vectos, every genera and even species have some differences in the behavior and incidence in human habitations. Therefore, the entomological component should be added and discussed. Wil the measures taken in this case will affect all the species/genera. Will be the same for all in the triatominae in the area? etc. Are there some species for which changes in construction may not affect that much?

We agree that different species/genera have different incidence in dwellings. We have added to the setting section the three species we have encountered I these communities.

We do not know the specific impact of this intervention on the three species we have encountered for likelihood of home (re)infestation following home reconstruction. Our anecdotal observation is that none of the eight homes that have been rebuilt have become infested. We are reluctant to report that finding here because systemic investigations of homes that have and have not been reconstructed is a project we intend to conduct in July 2023 and July 2025.

Reviewer 5 Report

Dear Authors,

The manuscript entitled "Identifying barriers and facilitators for home reconstruction for  prevention of Chagas Disease: An interview study in rural Loja  province, Ecuador" brings a very interesting approach on a strategy for the prevention of Chagas disease trough home reconstruction with the community participation in Ecuador. The study identifies the most relevant facilitators and barriers related to the success of this initiative. Overall the manuscript is adequately designed and well written, although some minor correction to give a more clarity are required.

1- Although book and paper texts commonly refer that "the disease is transmitted", what is in fact transmitted is the etiological agent, which causes the disease.

Line 42 - The disease is transmitted by the parasite Trypanosoma cruzi carried in feces... is totally wrong sentence. Please correct!!! Trypanosoma cruzi, the etiological agent of Chagas disease, is transmitted through the excreta of triatomine bugs...

2- Another issue refers to the mention of the names of the interviewees, who in my opinion should remain anonymous. I recommend consulting the guidelines of the human research ethics committee of your institutions and the journal itself.

Author Response

We thank the reviewer for their comments.

We report their original comments in plain text and our response in italics immediately following.

Dear Authors,

The manuscript entitled "Identifying barriers and facilitators for home reconstruction for  prevention of Chagas Disease: An interview study in rural Loja  province, Ecuador" brings a very interesting approach on a strategy for the prevention of Chagas disease trough home reconstruction with the community participation in Ecuador. The study identifies the most relevant facilitators and barriers related to the success of this initiative. Overall the manuscript is adequately designed and well written, although some minor correction to give a more clarity are required.

1- Although book and paper texts commonly refer that "the disease is transmitted", what is in fact transmitted is the etiological agent, which causes the disease.

We have re-written this sentence as advised below.

Line 42 - The disease is transmitted by the parasite Trypanosoma cruzi carried in feces... is totally wrong sentence. Please correct!!! Trypanosoma cruzi, the etiological agent of Chagas disease, is transmitted through the excreta of triatomine bugs...

We have changed this sentence as advised by the reviewer.

2- Another issue refers to the mention of the names of the interviewees, who in my opinion should remain anonymous. I recommend consulting the guidelines of the human research ethics committee of your institutions and the journal itself.

We agree that anonymity of participants is important. As stated on line 167-168, each participant was assigned a pseudonym. The names reported in the paper are not the actual names of the participants.

Round 2

Reviewer 1 Report

Dear authors,

I appreciate that the requested changes have been made. I have just a few new suggestions:

In line 42, the taxonomic classification for T. cruzi is still missing. It would also be important to include the author and year of description of the species, which should be presented after the name in the first mention of the species. This should also be included in the triatomine species included in lines 96-97.

Trypanosoma cruzi (Chagas, 1909)

Rhodnius ecuadoriensis Lent & León, 1958

Triatoma carrioni Larrousse, 1926

Panstrongylus chinai (Del Ponte, 1929)

Best Regards.

Author Response

Thank you for your comments.

As before, we report the reviewer comments in plain text and our response in italicized text.

In line 42, the taxonomic classification for T. cruzi is still missing. It would also be important to include the author and year of description of the species, which should be presented after the name in the first mention of the species. This should also be included in the triatomine species included in lines 96-97.

Trypanosoma cruzi (Chagas, 1909)

Rhodnius ecuadoriensis Lent & León, 1958

Triatoma carrioni Larrousse, 1926

Panstrongylus chinai (Del Ponte, 1929)

We have added the taxonomic classification for T. Cruzi (Kinetoplastida: Trypanosomatidae: Trypanosoma).

For all species, we have added author and year of description of the species as advised.

Reviewer 4 Report

The manuscript has been improved, still can be improved by reading and searching references on the species that are common to the area and then add a small paragraph discussion (even what the authors tell in the rebuttal could be fine as discussion!). It would also be good to include in the materials and methods what papers/guides were used to identify the triatomines spotted in the area. All the rest of the ms seems to be OK

Author Response

We thank the reviewer for these comments.

As before, we report the reviewer's comments in plain text and our reply in italics.

The manuscript has been improved, still can be improved by reading and searching references on the species that are common to the area and then add a small paragraph discussion (even what the authors tell in the rebuttal could be fine as discussion!). It would also be good to include in the materials and methods what papers/guides were used to identify the triatomines spotted in the area. All the rest of the ms seems to be OK

We are confused by the first comment about searching references on the species common to the area. In the previous round of review, we have referred to the previous papers that represent the most comprehensive investigation of the species common to this area. Specifically, reference 15 and 22 represent the most comprehensive investigation of domiciliary species of triatomine insects in the three communities. This appears around lines 95-98.

As advised by the reviewer, we have added to the discussion section . The rebuttal paragraph has been revised and incorporated into the discussion as advised by the reviewer. This can be found around lines 602-612.

We believe that the request to report materials and methods used to identify the triatomines spotted in the area are beyond the scope of this manuscript. The present manuscript did not discuss identification of the insect species. We believe the methods in 15 and 22 are described well in those papers. The “three species we have encountered in these communities” refers to the fact that authors on the present paper (Grijalva, Baus, and Nieto-Sanchez) were authors on those previous papers. To report those materials and method would (a) reproduce the work that has already been published In those papers and (b) make it appear as if the research reported in the present paper was about insect identification when the scope of the present paper is about the barriers and facilitators to home reconstruction. We have therefore not included this material.